# Enhancing Conversational Search: Large Language Model-Aided Informative Query Rewriting

**Fanghua Ye**
University College London
fanghua.ye.19@ucl.ac.uk

**Meng Fang**
University of Liverpool
Meng.Fang@liverpool.ac.uk

**Shenghui Li**
Uppsala University
shenghui.li@it.uu.se

**Emine Yilmaz**
University College London
emine.yilmaz@ucl.ac.uk

## Abstract

Query rewriting plays a vital role in enhancing conversational search by transforming context-dependent user queries into standalone forms. Existing approaches primarily leverage human-rewritten queries as labels to train query rewriting models. However, human rewrites may lack sufficient information for optimal retrieval performance. To overcome this limitation, we propose utilizing large language models (LLMs) as query rewriters, enabling the generation of informative query rewrites through well-designed instructions. We define four essential properties for well-formed rewrites and incorporate all of them into the instruction. In addition, we introduce the role of rewrite editors for LLMs when initial query rewrites are available, forming a "rewrite-then-edit" process. Furthermore, we propose distilling the rewriting capabilities of LLMs into smaller models to reduce rewriting latency. Our experimental evaluation on the QReCC dataset demonstrates that informative query rewrites can yield substantially improved retrieval performance compared to human rewrites, especially with sparse retrievers.[1]

## 1 Introduction

Conversational search has gained significant prominence in recent years with the proliferation of digital virtual assistants and chatbots, enabling users to engage in multiple rounds of interactions to obtain information (Radlinski and Craswell, 2017; Dalton et al., 2021; Gao et al., 2023). This emerging search paradigm offers remarkable advantages in assisting users with intricate information needs and complex tasks (Yu et al., 2021). However, a fundamental challenge in conversational search lies in accurately determining users' current search intents within the conversational context.

An effective approach that has gained increasing attention addresses this challenge of conversational

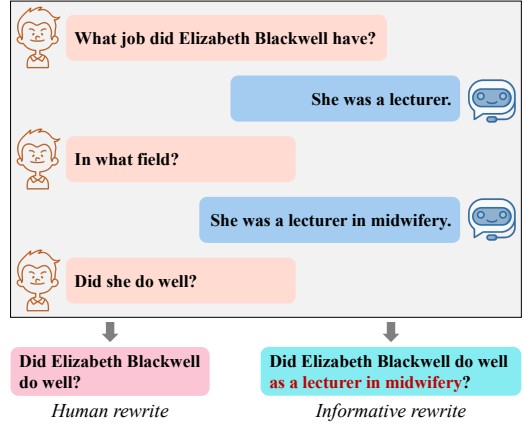

Figure 1: An example showing that human rewrites may overlook valuable contextual information. Specifically, the omission of the phrase "as a lecturer in midwifery" makes it challenging for retrieval systems to understand the original query comprehensively.

context modeling by performing *query rewriting* (Elgohary et al., 2019; Yu et al., 2020; Vakulenko et al., 2021; Wu et al., 2022; Mo et al., 2023). This approach transforms context-dependent user queries into self-contained queries, thereby allowing the utilization of existing *off-the-shelf* retrievers that have been extensively validated for standalone queries. For example, the user query "*Did she do well?*" illustrated in Figure 1 can be rewritten into "*Did Elizabeth Blackwell do well as a lecturer in midwifery?*" which is context-independent.

Previous studies (Anantha et al., 2021; Vakulenko et al., 2021; Qian and Dou, 2022; Hao et al., 2022) predominantly depend on human-rewritten queries as supervised labels to train query rewriting models. Although human-rewritten queries tend to perform better than the original queries, they may not be informative enough for optimal retrieval performance (Chen et al., 2022; Wu et al., 2022). This limitation arises from the fact that human rewriters are only concerned with addressing ambiguity issues, such as coreference and omission, when transforming the original query into a self-contained

---

[1]Our implementation is available at: https://github.com/smartyfh/InfoCQR.

form. Such a simple rewriting strategy may overlook lots of valuable information within the conversational context (refer to Figure 1 for an example), which has the potential to enhance the effectiveness of the retriever. As a consequence, existing query rewriting models learned from human rewrites can only achieve sub-optimal performance.

A straightforward approach to improving the informativeness of rewritten queries is to provide human annotators with more comprehensive instructions so that they can rewrite the original queries to be not only unambiguous but also informative. However, this approach has several disadvantages, including being expensive, increasing workload for human annotators, and potentially leading to higher inconsistencies among rewrites from different annotators. Therefore, it is necessary to explore alternative approaches.

In this paper, we propose the utilization of large language models (LLMs) for query rewriting, leveraging their impressive capabilities in following instructions and demonstrations (Brown et al., 2020; Wei et al., 2021; Ouyang et al., 2022; Wei et al., 2023). We consider two settings to prompt LLMs as query rewriters. In the zero-shot learning setting, only an instruction is provided, while in the few-shot learning setting, both an instruction and a few demonstrations are given. To develop suitable instructions, we first identify four essential properties that characterize a well-formed rewritten query. Then, we design an instruction that incorporates all four properties. However, generating rewrites with all these properties may pose challenges for LLMs due to the intricacy of the instruction (Ouyang et al., 2022; Jang et al., 2023). In view of this, we propose an additional role for LLMs as rewrite editors. Inspired by the fact that humans excel at editing rather than creating from scratch, the purpose of the rewrite editor is to edit initial rewrites provided, forming a "rewrite-then-edit" process. These initial rewrites can be generated by smaller query rewriting models or even by the LLM itself. Furthermore, considering the potential time overhead and high costs associated with LLMs, we suggest distilling their rewriting capabilities into smaller models using their generated rewrites as training labels.

Our contributions are summarized as follows:

- We are the first to introduce the concept of informative conversational query rewriting and meticulously identify four desirable properties that a well-crafted rewritten query should possess.

- We propose to prompt LLMs as both query rewriters and rewrite editors by providing clear instructions that incorporate all the desirable properties. In addition, we employ distillation techniques to condense the rewriting capabilities of LLMs into smaller models to improve rewriting efficiency.

- We demonstrate the effectiveness of informative query rewriting with two off-the-shelf retrievers (sparse and dense) on the QReCC dataset. Our results show that informative query rewrites can outperform human rewrites, particularly in the context of sparse retrieval.

## 2 Task Formulation

The primary objective of conversational search is to identify relevant passages from a vast collection of passages in response to the current user query. Formally, let $Q_i$ and $A_i$ be the user query and system response at turn $i$, respectively. Furthermore, let $\mathcal{X}_t = \{Q_1, A_1, \ldots, Q_{t-1}, A_{t-1}\}$ represent the conversational context up to turn $t$. Then, the task of conversational search can be formulated as retrieving top-$k$ relevant passages, denoted as $\mathcal{R}_k$, from a large passage collection $\mathcal{C}$ given the current user query $Q_t$ and its associated context $\mathcal{X}_t$. This retrieval process is accomplished by a retriever defined as $f : (Q_t, \mathcal{X}_t, \mathcal{C}) \rightarrow \mathcal{R}_k$, where $\mathcal{R}_k$ is a subset of $\mathcal{C}$ and $k$ is considerably smaller than the total number of passages in $\mathcal{C}$.

The unique challenge in conversational search is incorporating the conversational context while retrieving relevant passages, which cannot be directly addressed by existing retrievers designed for standalone queries. Additionally, re-training retrievers tailored for conversational queries can be expensive or even infeasible due to complex system designs or limited data availability (Wu et al., 2022). To overcome the need for re-training, query rewriting is employed as an effective solution (Lin et al., 2021c; Mo et al., 2023). Query rewriting involves transforming the context-dependent user query $Q_t$ into a self-contained standalone query $Q'_t$ by extracting relevant information from the context $\mathcal{X}_t$. Consequently, any existing off-the-shelf retrieval systems designed for standalone queries can be leveraged by taking $Q'_t$ as the input query to find passages that are relevant to the original user query $Q_t$, i.e., $f : (Q'_t, \mathcal{C}) \rightarrow \mathcal{R}_k$.

The utilization of query rewriting shifts the challenge of modeling conversational context from the retriever end to the query rewriting model end. As

Figure 2: Our proposed approach involves prompting LLMs as query rewriters and rewrite editors through clear and well-designed instructions, along with appropriate demonstrations. In the absence of demonstrations, the LLM functions as a zero-shot query rewriter. We explicitly incorporate the requirement that rewritten queries should be as informative as possible into the instructions for generating informative query rewrites.

thus, the effectiveness of retrieval results heavily relies on the employed query rewriting models. Only when appropriate rewritten queries are generated can an off-the-shelf retrieval system return highly relevant passages.

## 3 Approach

In contrast to relying on human annotators to generate more informative rewrites or developing more complex models to closely replicate existing human rewrites, we propose to prompt LLMs to generate informative query rewrites simply by providing clear instructions and appropriate demonstrations, avoiding the requirement for extensive human effort and complicated model designs. Figure 2 illustrates our proposed approach.

### 3.1 Prompting LLM as Query Rewriter

Recent work (Wei et al., 2021; Ouyang et al., 2022; Peng et al., 2023) has demonstrated the strong capability of LLMs in following given instructions to generate coherent and contextually appropriate text. Inspired by this, it is natural to consider employing LLMs as query rewriters. Before delving into the details of how we can prompt an LLM as a query rewriter, we first describe the desirable properties

that a well-crafted rewritten query should possess:

- **Correctness:** The rewritten query should preserve the meaning of the original query, ensuring that the user's intent remains unchanged.

- **Clarity:** The rewritten query should be unambiguous and independent of the conversational context, enabling it to be comprehensible by people outside the conversational context. This clarity can be achieved by addressing coreference and omission issues arising in the original query.

- **Informativeness:** The rewritten query should incorporate as much valuable and relevant information from the conversational context as possible, thereby providing more useful information to the off-the-shelf retriever.

- **Nonredundancy:** The rewritten query should avoid duplicating any query previously raised in the conversational context, as it is important to ensure that the rewritten query only conveys the intent and meaning of the current query.

In order to effectively instruct an LLM in generating query rewrites that embody the aforementioned four properties, it is essential to formulate

appropriate instructions. As an illustrative example, we adopt the following instruction in this work:

> "*Given a question and its context, decontextualize the question by addressing coreference and omission issues. The resulting question should retain its original meaning and be as informative as possible, and should not duplicate any previously asked questions in the context.*"[2]

This instruction takes all four desirable properties of a good rewritten query into account simultaneously. Building upon this instruction, we explore two settings to prompt an LLM for query rewriting.

### 3.1.1 Zero-Shot Learning (ZSL) Setting

In the ZSL setting, the LLM is instructed to generate a rewritten query $Q'_t$ using only the information provided by the current query $Q_t$ and its associated conversational context $\mathcal{X}_t$, without having access to any human-labeled instances. In this setting, we entirely rely on the LLM's capability to understand and follow instructions to perform query rewriting. Specifically, we append $\mathcal{X}_t$ and $Q_t$ to the instruction $I$ as the prompt and feed this prompt to the LLM for sampling the rewrite $Q'_t$:

$$Q'_t \sim \mathcal{LLM}(I||\mathcal{X}_t||Q_t), \qquad (1)$$

where $||$ denotes concatenation. The detailed format of the prompt is shown in Appendix D.

### 3.1.2 Few-Shot Learning (FSL) Setting

In the FSL setting, the LLM is provided with both the instruction and a small number of demonstrations. This type of prompting is commonly referred to as in-context learning, which has been shown to be effective in adapting LLMs to new tasks (Brown et al., 2020; Min et al., 2022a,b; Wei et al., 2023; Sun et al., 2023; Ram et al., 2023). In this setting, each demonstration consists of a query $Q$, a conversational context $\mathcal{X}$, and a rewrite $Q'$. We denote the concatenation of these demonstrations as:

$$\mathcal{D} = (\mathcal{X}^1, Q^1, Q'^1)||\dots||(\mathcal{X}^n, Q^n, Q'^n), \quad (2)$$

where $n$ represents the total number of demonstrations. By placing $\mathcal{D}$ between the instruction $I$ and

the test instance $(\mathcal{X}_t, Q_t)$ as the prompt to the LLM, the rewrite $Q'_t$ is then sampled as follows:

$$Q'_t \sim \mathcal{LLM}(I||\mathcal{D}||\mathcal{X}_t||Q_t). \qquad (3)$$

Note that the query rewrites utilized in the demonstrations should be well-designed, ensuring that they have the aforementioned four properties. Otherwise, the LLM may be misled by these demonstrations. For a more detailed description of the demonstrations used in our experiments, please refer to Appendix D.

### 3.2 Prompting LLM as Rewrite Editor

Despite the proficiency of LLMs in following instructions and demonstrations, recent work (Dong et al., 2022; Liu et al., 2022; Mosbach et al., 2023) suggests that they may encounter difficulties when faced with complex tasks or intricate requirements. This limitation highlights that it can be challenging for LLMs to generate query rewrites with all the desirable properties mentioned above. In order to address this challenge, we propose an alternative approach in which an LLM is prompted as a rewrite editor whose primary function is to edit provided initial rewrites instead of being prompted as a query rewriter who needs to generate query rewrites from scratch. This approach draws inspiration from the observation that humans often find it easier to edit existing content than to create it from scratch.

In this work, we adopt the FSL setting to prompt LLMs as rewrite editors. In addition to the query $Q$, the conversational context $\mathcal{X}$, and the rewrite $Q'$, we introduce an initial rewrite $\hat{Q}$ for each demonstration. We represent the concatenation of these augmented demonstrations as:

$$\tilde{\mathcal{D}} = (\mathcal{X}^1, Q^1, \hat{Q}^1, Q'^1)||\dots||(\mathcal{X}^n, Q^n, \hat{Q}^n, Q'^n). \qquad (4)$$

For a test instance $(\mathcal{X}_t, Q_t)$, accompanied by an initial rewrite $\hat{Q}_t$, we obtain the edited (final) rewrite $Q'_t$ through the following procedure:

$$Q'_t \sim \mathcal{LLM}(\tilde{I}||\tilde{\mathcal{D}}||\mathcal{X}_t||Q_t||\hat{Q}_t), \qquad (5)$$

where $\tilde{I}$ denotes the modified instruction. Please refer to Figure 2 and Appendix D for details.

The initial rewrite can be generated by a small query rewriting model, such as T5QR (Lin et al., 2020; Wu et al., 2022). It can also be generated by an LLM, following the prompting method described in the previous subsection. When an LLM is employed as both the query rewriter and rewrite editor, the "*rewrite-then-edit*" process enables the LLM to perform self-correction (Gou et al., 2023).

---

[2]Note that in this instruction, we use the term "question" instead of "query", as each query is referred to as a question in the dataset we employed for experimentation. We believe this tiny variation would not significantly impact the quality of generated rewrites. Additionally, other instructions with a similar meaning can also be applied.

### 3.3 Distillation: LLM as Rewriting Teacher

One major obstacle in effectively leveraging LLMs for query rewriting is the substantial demand for memory and computational resources (Hsieh et al., 2023), which can further result in significant time overhead. Besides, the cost can be extremely high when there is a lack of in-house models, necessitating the reliance on third-party API services as the only option. To address these issues, we propose to fine-tune a small query rewriting model using rewrites generated by an LLM as ground-truth labels. In this approach, the LLM assumes the role of a teacher, while the smaller query rewriting model acts as a student. The fine-tuning process distills the teacher's rewriting capabilities into the student. This technique is known as knowledge distillation (Gou et al., 2021) and has recently been utilized to distill LLMs for various other tasks (Shridhar et al., 2022; Magister et al., 2022; Marjieh et al., 2023).

Following previous work (Lin et al., 2020; Wu et al., 2022), we adopt T5 (Raffel et al., 2020) as the student model (i.e., the small query rewriting model). The input to the model is the concatenation of all utterances in the conversational context $\mathcal{X}_t$ and the current user query $Q_t$. In order to differentiate between user queries and system responses, we prepend a special token <Que> to each user query and a special token <Ans> to each system response. The output of the model is the rewrite $Q'_t$, which is sampled from the employed LLM. The model is fine-tuned using the standard cross-entropy loss to maximize the likelihood of generating $Q'_t$.

## 4 Experimental Setup

### 4.1 Dataset & Evaluation Metrics

Following previous work (Wu et al., 2022; Mo et al., 2023), we leverage QReCC (Anantha et al., 2021) as our experimental dataset. QReCC consists of 14K open-domain English conversations with a total of 80K question-answer pairs. Each user question is accompanied by a human-rewritten query, and the answers to questions within the same conversation may be distributed across multiple web pages. There are in total 10M web pages with each divided into several passages, leading to a collection of 54M passages[3]. The task of conversational search is to find relevant passages for each question from this large collection and gold passage labels are provided if any. The conversations in QReCC

are sourced from three existing datasets, including QuAC (Choi et al., 2018), Natural Questions (Kwiatkowski et al., 2019), and TREC CAsT-19 (Dalton et al., 2020). For ease of differentiation, we refer to these subsets as *QuAC-Conv*, *NQ-Conv*, and *TREC-Conv*, respectively. Note that TREC-Conv only appears in the test set. For a comprehensive evaluation, we present experimental results not only on the overall dataset but also on each subset. For additional information and statistics regarding the dataset, please refer to Appendix A.

To evaluate the retrieval results, we adopt mean reciprocal rank (**MRR**), mean average precision (**MAP**), and Recall@10 (**R@10**) as evaluation metrics. We employ the pytrec_eval toolkit (Van Gysel and de Rijke, 2018) for the computation of all metric values.

### 4.2 Comparison Methods

Since our focus is on the effectiveness of informative query rewrites, two straightforward baseline methods are **Original**, which uses the user question in its original form as the search query, and **Human**, which utilizes the query rewritten by a human as the search query. We also include three supervised models as baselines, including **T5QR** (Lin et al., 2020), which fine-tunes the T5-base model (Raffel et al., 2020) as a seq2seq query rewriter, **ConQRR** (Wu et al., 2022), which employs reinforcement learning to train query rewriting models by directly optimizing retrieval performance, and **ConvGQR** (Mo et al., 2023), which combines query rewriting with potential answer generation to improve the informativeness of the search query.

For our proposed approach, we investigate four variants, namely *RW(ZSL)*, *RW(FSL)*, *ED(Self)*, and *ED(T5QR)*. **RW(ZSL)** prompts an LLM as a query rewriter in the ZSL setting, while **RW(FSL)** prompts an LLM as a query rewriter in the FSL setting. By comparison, **ED(Self)** prompts an LLM as a rewrite editor, wherein the initial rewrites are generated by RW(FSL) with the same LLM applied. **ED(T5QR)** also prompts an LLM as a rewrite editor, but the initial rewrites are generated by T5QR. For simplicity, we only prompt LLMs as rewrite editors in the FSL setting.

### 4.3 Retrieval Systems

We experiment with two types of off-the-shelf retrievers to explore the effects of informativeness in query rewrites on conversational search:

---

[3]The dataset and passage collection are available at https://zenodo.org/record/5115890#.YZ8kab3MI-Q.

| | Query | QReCC (8209) | | | QuAC-Conv (6396) | | | NQ-Conv (1442) | | | TREC-Conv (371) | | |
|---|---|---|---|---|---|---|---|---|---|---|---|---|---|
| | | MRR | MAP | R@10 | MRR | MAP | R@10 | MRR | MAP | R@10 | MRR | MAP | R@10 |
| **Sparse (BM25)** | Original | 9.30 | 8.87 | 15.50 | 9.29 | 8.84 | 15.20 | 9.06 | 8.64 | 15.14 | 10.30 | 10.27 | 22.10 |
| | Human | 39.81 | 38.45 | 62.65 | 40.32 | 38.98 | 62.90 | 40.78 | 39.05 | **63.80** | 27.34 | 27.04 | **53.77** |
| | T5QR | 33.67 | 32.50 | 53.68 | 34.04 | 32.90 | 53.83 | 34.24 | 32.66 | 53.92 | 25.23 | 24.96 | 50.13 |
| | ConQRR | 38.30 | - | 60.10 | 39.50 | - | 61.60 | 37.80 | - | 58.00 | 19.80 | - | 43.50 |
| | ConvGQR | 44.10 | - | 64.40 | - | - | - | - | - | - | - | - | - |
| | RW(ZSL) | 42.63 | 41.31 | 60.46 | 45.43 | 44.11 | 63.20 | 36.43 | 34.81 | 54.69 | 18.50 | 18.26 | 35.58 |
| | RW(FSL) | 46.96 | 45.53 | 65.57 | 49.81 | 48.38 | 68.28 | 41.51 | 39.71 | 60.13 | 19.02 | 18.86 | 39.89 |
| | ED(Self) | **49.39** | **47.89** | **67.01** | **53.01** | **51.52** | **70.46** | 41.57 | 39.69 | 59.63 | 17.43 | 17.08 | 36.25 |
| | ED(T5QR) | 47.93 | 46.40 | 66.25 | 50.67 | 49.18 | 68.84 | **42.69** | **40.64** | 60.67 | 21.04 | 20.79 | 43.26 |
| **Dense (GTR)** | Original | 12.12 | 11.49 | 18.74 | 11.34 | 10.69 | 17.79 | 13.11 | 12.57 | 19.49 | 21.76 | 21.11 | 32.08 |
| | Human | 43.15 | 41.27 | 66.12 | 40.67 | 38.92 | 64.59 | **54.01** | **51.25** | 73.13 | 43.74 | 42.98 | **65.23** |
| | T5QR | 37.67 | 35.93 | 58.65 | 35.51 | 33.88 | 57.23 | 46.95 | 44.47 | 64.43 | 38.94 | 38.16 | 60.51 |
| | ConQRR | 41.80 | - | 65.10 | 41.60 | - | 65.90 | 45.30 | - | 64.10 | 32.70 | - | 55.20 |
| | ConvGQR | 42.00 | - | 63.50 | - | - | - | - | - | - | - | - | - |
| | RW(ZSL) | 40.64 | 38.95 | 62.28 | 40.12 | 38.48 | 62.47 | 44.85 | 42.57 | 63.58 | 33.26 | 32.88 | 54.09 |
| | RW(FSL) | 43.89 | 42.09 | 66.45 | 43.50 | 41.78 | 66.87 | 48.60 | 46.12 | 68.10 | 32.37 | 31.79 | 52.65 |
| | ED(Self) | **44.99** | **43.19** | **67.34** | **45.21** | **43.48** | **68.30** | 47.64 | 45.20 | 67.27 | 30.91 | 30.48 | 51.03 |
| | ED(T5QR) | 44.76 | 42.90 | 66.64 | 44.29 | 42.50 | 66.65 | 49.67 | 47.12 | 69.22 | 33.90 | 33.43 | 56.47 |

Table 1: Passage retrieval performance of sparse and dense retrievers with various query rewriting methods on the QReCC test set and its three subsets. The best results are shown in bold and the second-best results are underlined.

**BM25** BM25 (Robertson et al., 2009) is a classic sparse retriever. Following Anantha et al. (2021), we employ Pyserini (Lin et al., 2021a) with hyparameters $k1 = 0.82$ and $b = 0.68$.

**GTR** GTR (Ni et al., 2022) is a recently proposed dense retriever[4]. It has a shared dual-encoder architecture and achieves state-of-the-art performance on multiple retrieval benchmarks.

### 4.4 Implementation Details

We adopt ChatGPT (gpt-3.5-turbo) provided by OpenAI through their official API[5] as the LLM in our experiments. During inference, we use greedy decoding with a temperature of 0. In the FSL setting, we utilize four demonstrations (i.e., $n = 4$). We employ Pyserini (Lin et al., 2021a) for sparse retrieval and Faiss (Johnson et al., 2019) for dense retrieval. For each user query, we retrieve 100 passages (i.e., $k = 100$). We disregard test instances without valid gold passage labels. As a result, we have 8209 test instances in total, with 6396, 1442, and 371 test instances for QuAC-Conv, NQ-Conv, and TREC-Conv, respectively. For more implementation details, please refer to Appendix B.

---

[4]We use the T5-base version https://huggingface.co/sentence-transformers/gtr-t5-base.

[5]platform.openai.com/docs/api-reference/chat

## 5 Experimental Results

### 5.1 Main Results

Table 1 presents the retrieval performance of different query rewriting methods on the QReCC test set and its subsets. Our key findings are summarized as follows. **(I)** All query rewriting methods outperform the original query, validating the importance of query rewriting. **(II)** Our approaches, ED(Self) and ED(T5QR), consistently achieve the best and second-best results on the overall QReCC test set. Notably, they both surpass human rewrites. For example, ED(Self) demonstrates a substantial absolute improvement of 9.58 in MRR scores for sparse retrieval compared to human rewrites. RW(FSL) also performs better than human rewrites, while RW(ZSL) fails to show consistent improvements over human rewrites. These results emphasize the value of informative query rewriting and in-context demonstrations. **(III)** The supervised models, T5QR and ConQRR, exhibit worse performance than human rewrites, suggesting that relying solely on learning from human rewrites leads to sub-optimal results. Although ConvGQR beats human rewrites in sparse retrieval, its performance gain mainly derives from generated potential answers rather than more informative query rewrites. **(IV)** Dense retrieval improvements are

| Query | QuAC-Conv | | NQ-Conv | | TREC-Conv | |
|---|---|---|---|---|---|---|
| | AT | %OT | AT | %OT | AT | %OT |
| Original | 6.75 | 61.23 | 6.31 | 64.41 | 6.01 | 74.76 |
| T5QR | 9.80 | 83.63 | 8.67 | 85.27 | 7.31 | 90.32 |
| RW(ZSL) | 16.51 | 68.71 | 14.48 | 73.55 | 12.53 | 65.03 |
| RW(FSL) | 16.95 | 76.66 | 15.74 | 81.68 | 15.07 | 74.11 |
| ED(Self) | 22.00 | 78.88 | 19.82 | 82.99 | 21.85 | 76.22 |
| ED(T5QR) | 17.93 | 85.16 | 15.08 | 89.28 | 15.66 | 87.78 |
| Human | 10.76 | 100 | 8.98 | 100 | 7.35 | 100 |

Table 2: Average number of tokens (AT) and the percentage of overlapping tokens (OT) with human rewrites in queries produced by different rewriting methods.

| | Query | MRR | MAP | R@10 |
|---|---|---|---|---|
| **Sparse (BM25)** | Human | 39.81 | 38.45 | 62.65 |
| | RW(ZSL) | 42.63 | 41.31 | 60.46 |
| | $\widehat{RW}$(ZSL) | 41.52 | 40.20 | 59.60 |
| **Dense (GTR)** | Human | 43.15 | 41.27 | 66.12 |
| | RW(ZSL) | 40.64 | 38.95 | 62.28 |
| | $\widehat{RW}$(ZSL) | 39.94 | 38.28 | 61.44 |

Table 3: Ablation study by removing the informativeness requirement from the instruction in RW(ZSL).

less effective than sparse retrieval. For example, ED(Self) only outperforms human rewrites by 1.84 MRR scores when using the dense retriever GTR. This discrepancy arises due to the need for domain-specific passage and query encoders in dense retrieval. In our experiments, the GTR model is kept fixed without fine-tuning, which limits the full potential of dense retrieval. Besides, ConvGQR also shows inferior dense retrieval performance, further indicating that a fixed general dense retriever cannot fully demonstrate the superiority of informative query rewrites. **(V)** Breaking down the results by subsets reveals that our proposed approaches can consistently achieve higher performance on the QuAC-Conv subset. They also prevail in terms of MRR and MAP for sparse retrieval and achieve the second-best results for dense retrieval on the NQ-Conv subset. However, our approaches are inferior to human rewrites and T5QR on the TREC-Conv subset. One reason is that TREC-Conv contains many obscure questions, making it challenging for an LLM to accurately understand true user needs. It is seen that even human rewrites perform worse on TREC-Conv than on QuAC-Conv and NQ-Conv in sparse retrieval. Additionally, the questions in TREC-Conv are more self-contained and require less rewriting, as evidenced by higher ROUGE-1 scores (Lin, 2004) between human rewrites and original questions compared to QuAC-Conv and NQ-Conv. Specifically, the ROUGE-1 scores on TREC-Conv, QuAC-Conv, and NQ-Conv are 80.60, 69.73, and 72.16, respectively. **(VI)** Both ED(Self) and ED(T5QR) outperform RW(FSL), showing the significance of prompting LLMs as rewrite editors. While ED(T5QR) performs worse than ED(Self) on the overall QReCC test set, it excels on the NQ-Conv and TREC-Conv subsets, benefiting from the fact that T5QR is trained with human rewrites.

In summary, this study confirms the importance

of informative query rewriting and the effectiveness of our proposed approaches of prompting LLMs as query rewriters and rewrite editors. The study also suggests that it is crucial to take query characteristics into account when performing query rewriting with LLMs, which we leave as our future work.

## 5.2 Quantitative Analyses of Query Rewrites

The previous results have demonstrated the effectiveness of informative query rewrites generated by our proposed approaches in enhancing conversational search. To gain more insights into the quality of these rewrites, we employ the average number of tokens per rewrite as a measurement of informativeness and the percentage of tokens in human rewrites that also appear in the generated rewrites as a measurement of correctness. We assume that a rewrite containing more tokens from the corresponding human rewrite is more likely to be correct. The results are shown in Table 2. We observe that our proposed approaches consistently generate longer rewrites than human rewrites, with ED(Self) producing the longest rewrites overall. This implies that the rewrites generated by our proposed approaches are more informative. We also observe that T5QR generates shorter rewrites than human rewrites, indicating that relying solely on learning from human rewrites fails to produce informative rewrites. Moreover, our proposed approaches, albeit without supervised fine-tuning, achieve relatively high correctness compared to human rewrites. For example, more than 76% of tokens in human rewrites are included in the rewrites generated by ED(Self). ED(T5QR) even exhibits higher correctness than T5QR on the QuAC-Conv and NQ-Conv subsets. Finally, the longer rewrites and higher percentage of shared tokens with human rewrites (except RW on TREC-Conv), compared to the original queries, suggest to some extent that the rewrites generated by our approaches have fair clarity.

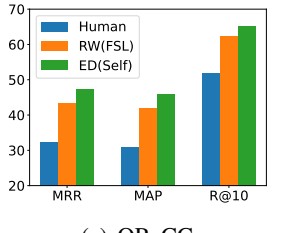 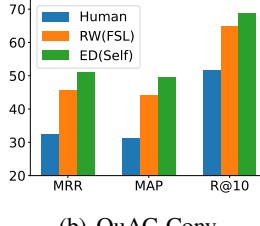 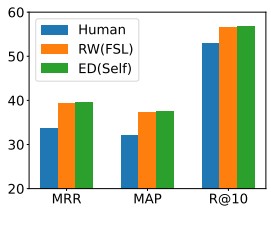 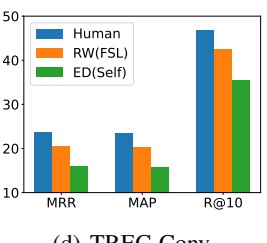

| (a) QReCC | (b) QuAC-Conv | (c) NQ-Conv | (d) TREC-Conv |

Figure 3: Distillation results of using BM25 as the retriever. The legends indicate the sources of fine-tuning labels.

## 5.3 Ablation Study

We conduct an ablation study by removing the informativeness requirement from the instruction utilized by RW(ZSL) (i.e., removing the phrase "*and be as informative as possible*"), resulting in a modified version denoted as $\widehat{\text{RW}}$(ZSL). Table 3 reports the results. We find that for both sparse and dense retrieval, $\widehat{\text{RW}}$(ZSL) achieves lower performance across all three evaluation metrics than RW(ZSL), demonstrating that it is valuable to incorporate informativeness requirement into the instruction for generating informative query rewrites. Interestingly, $\widehat{\text{RW}}$(ZSL) outperforms human rewrites in terms of MRR and MAP for sparse retrieval, which again verifies the notion that human rewrites may fail to yield optimal retrieval performance. See Appendix C.3 for ablation results of the other three desirable properties.

## 5.4 Distillation Results

Figure 3 shows the distillation results using BM25 as the retriever. In this study, we sample 10K training instances and employ RW(FSL) and ED(Self) to generate labels for fine-tuning the T5QR model. For comparison, we include the results with human rewrites as training labels. We find that distillation outperforms using human rewrites as labels on the QReCC test set. Notably, distillation with only 10K training instances can achieve superior results than directly utilizing human rewrites as search queries in terms of MRR and MAP. On the QuAC-Conv and NQ-Conv subsets, distillation also consistently demonstrates improved performance. However, for TREC-Conv, fine-tuning with human rewrites leads to better outcomes. Distillation not only improves retrieval performance but also reduces time overhead. See Appendix C.4 for latency analyses.

## 6 Related Work

Conversational search addresses users' information needs through iterative interactions (Radlinski

and Craswell, 2017; Rosset et al., 2020). It allows users to provide and seek clarifications (Xu et al., 2019) and explore multiple aspects of a topic, thereby excelling at fulfilling complex information needs. The primary challenge in conversational search is accurately identifying users' search intents from their contextualized and potentially ambiguous queries (Ye et al., 2022a; Keyvan and Huang, 2022; Ye et al., 2022b; Wang et al., 2023; Owoicho et al., 2023; Zhu et al., 2023).

Most existing work (Yu et al., 2021; Lin et al., 2021b; Kim and Kim, 2022; Li et al., 2022a; Mao et al., 2022) addresses this challenge by regarding the concatenation of the current user query with its associated conversational context as a standalone query. However, using this concatenation directly as input to search systems could result in poor retrieval performance (Lin et al., 2021b). Moreover, this approach requires training specialized retrievers such as dual encoders (Karpukhin et al., 2020; Xiong et al., 2020; Khattab and Zaharia, 2020), which can be challenging or even impractical in many real-world scenarios (Wu et al., 2022).

Another line of research addresses this challenge through query rewriting (Elgohary et al., 2019; Wu et al., 2022; Qian and Dou, 2022; Yuan et al., 2022; Li et al., 2022b; Mo et al., 2023), which converts the original query into a standalone one. However, these approaches mainly rely on human rewrites to train query rewriting models. As shown in our experiments, human rewrites may lack sufficient informativeness, hence leading to sub-optimal performance of these rewriting models.

Alternatively, some studies employ query expansion to address this challenge. They select relevant terms from the conversational context (Voskarides et al., 2020; Kumar and Callan, 2020) or generate potential answers (Mo et al., 2023) to augment the original query. The latter can be seamlessly integrated into our approach to leverage the knowledge within LLMs. We leave this study as future work.

# 7 Conclusion

In this work, we propose to prompt LLMs as query rewriters and rewrite editors for informative query rewrite generation. We are the first to introduce the concept of informative query rewriting and identify four properties that characterize a well-formed rewrite. We also propose distilling the rewriting capabilities of LLMs into smaller models to improve efficiency. Our experiments verify the significance of informativeness in query rewrites and the effectiveness of using LLMs for generating rewrites.

Despite the superb performance achieved by our proposed approach, there are multiple future directions that are worthy of exploration. For example, we can train an auxiliary model to decide whether queries generated by prompting LLMs should be preferred to those generated by models that have been finetuned on human rewrites. We can also combine human rewrites and LLM rewrites as labels through a proper weighting strategy to finetune query rewriting models. Furthermore, in this work, we have used a fixed set of demonstrations for all test queries. To achieve the best performance, it is essential to find appropriate demonstrations for each specific query. This would be an effective solution to tackle obscure or complicated queries. Another future direction can be parameter-efficient fine-tuning (e.g., LoRA (Hu et al., 2021)) of LLMs with retrieval performance as feedback. In this way, we will aim to optimize the helpfulness of rewritten queries rather than informativeness.

## Limitations

We identify three limitations of our proposed approach. Firstly, the utilization of LLMs as query rewriters and rewrite editors inevitably suffers from the shortcomings associated with LLMs. Our experiments indicate that LLMs do not always follow provided instructions, resulting in generated rewrites that fail to possess the four desirable properties. For example, these rewrites may contain duplicate questions from the conversational context, thereby violating the nonredundancy requirement. In Appendix C.5, we present a case study demonstrating that the original user query may even be misinterpreted, leading to incorrect query rewrites.

Secondly, although our experimental results have demonstrated improved retrieval performance, it is essential to emphasize that the effectiveness of informative query rewriting is highly dependent on the formatting of the passage collection. In scenar-

ios where passages are relatively short, the introduction of more information in the search query may have a detrimental effect, as it becomes more challenging for retrieval systems to determine the most relevant passages. Conversely, informative query rewriting should prove beneficial in the context of long passages or document retrieval.

Thirdly, in this work, we have experimented with only one LLM, namely ChatGPT, and therefore our findings may be biased toward this specific model. It is unclear if other LLMs can achieve the same level of performance. Further investigation with more LLMs is worthwhile.

## Ethics Statement

Query rewriting plays a crucial role as an intermediary process in conversational search, facilitating a clearer comprehension of user search intents. This process is beneficial in generating appropriate responses to users. The effectiveness of this approach can be further enhanced through informative query rewriting, resulting in the retrieval of more relevant passages. Nevertheless, it is important to acknowledge that our proposed approaches are subject to the inherent limitations of LLMs, such as hallucinations, biases, and toxicity. It is also important to filter out passages that contain offensive text from the passage collection to ensure reliable retrieval results when applying our proposed approaches in practical scenarios.

## Acknowledgements

This work was funded by the EPSRC Fellowship titled "Task Based Information Retrieval" (grant reference number EP/P024289/1) and the Alan Turing Institute.

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

## A   Additional Dataset Details

The original QReCC dataset is only divided into a training set and a test set. Following Kim and Kim (2022), we sample 2K conversations from the training set as the development set. The statistics are summarized in Table 4. To ensure that the first user question is always self-contained and unambiguous, we replace all first questions with their corresponding human rewrites. This pre-processing aligns with previous work (Anantha et al., 2021; Wu et al., 2022). We note that some questions in the test set lack valid gold passage labels. As these questions are irrelevant for retrieval evaluation, we remove them from the test set. Consequently, our final test set consists of 8,209 questions, with 6,396 for QuAC-Conv, 1,442 for NQ-Conv, and 371 for TREC-Conv. This filtering also helps reduce costs associated with using OpenAI APIs.

|           |     | Train  | Dev    | Test   |
|-----------|-----|--------|--------|--------|
| **QReCC** | #C  | 8,823  | 2,000  | 2,775  |
|           | #Q  | 51,928 | 11,573 | 16,451 |
| **QuAC-Conv** | #C | 6,008 | 1,300 | 1,816 |
|           | #Q  | 41,395 | 8,965  | 12,389 |
| **NQ-Conv** | #C | 2,815 | 700    | 879    |
|           | #Q  | 10,533 | 2,608  | 3,314  |
| **TREC-Conv** | #C | 0    | 0      | 80     |
|           | #Q  | 0      | 0      | 748    |

Table 4: Statistics of the QReCC dataset and its three subsets. #C represents the number of conversations and #Q denotes the number of questions.

## B   Additional Implementation Details

Throughout our experiments, we leverage ChatGPT as the LLM and set the maximum number of generation tokens to 2,560 for all four variants of our proposed approach, namely RW(ZSL), RW(FSL), ED(Self), and ED(T5QR). ED(Self) utilizes the rewrites generated by RW(FSL) as initial rewrites, while ED(T5QR) takes the rewrites produced by T5QR as initial results. For training, we initialize T5QR with the t5-base checkpoint from Hugging-Face[6] and select the best model based on the highest BLEU score (Papineni et al., 2002) with human rewrites on the development set. The training process involves 10 epochs with a batch size of 16 and

---

[6] huggingface.co/docs/transformers/model_doc/t5

gradient accumulation steps of 2 (i.e., an overall batch size of 32). We employ AdamW (Loshchilov and Hutter, 2017) as the optimizer and creat a linear schedule with warmup to adjust the learning rate dynamically. The peak learning rate is set to 1e-5, and the warmup ratio is 0.1. The maximum conversational context length is restricted to 384, and the maximum output length is set to 64. We use a fixed random seed of 42 and conduct experiments on a single TITAN RTX GPU card with 24GB memory. Greedy decoding is used for inference.

To conduct distillation, we first sample 10K questions from the training set and 2K questions from the development set. We then apply RW(FSL) and ED(Self) to generate rewrites as pseudo labels for training the T5QR model. In this experiment, the training settings remain the same as the previous one, except that the number of gradient accumulation steps is set to 1 (i.e., an overall batch size of 16) and the BLEU score is calculated based on the generated pseudo labels rather than human rewrites. We perform testing on the full test set.

We use Pyserini (Lin et al., 2021a) to construct the sparse index for the BM25 retrieval model, with default hyparameters of $k1 = 0.82$ and $b = 0.68$. These values were chosen according to the retrieval performance on MS MARCO (Bajaj et al., 2016), a non-conversational retrieval dataset. We adopt Faiss (Johnson et al., 2019) to build the dense index for the GTR retrieval model. When encoding queries and passages, the maximum length is set to 384 and the dimension of embedding vectors is 768. We utilize cosine similarity between a query vector and a passage vector to estimate their relevance. Building the dense index for 54M passages requires around 320GB of RAM. To efficiently handle this, we split the passage collection into 8 parts, conduct retrieval on each part, and subsequently merge the results.

## C   Additional Experimental Results

### C.1   Pairwise Comparison of Query Rewriting Methods

We provide a more in-depth analysis of different query rewriting methods by comparing the performance of every two methods on each instance within the QReCC test set. We leverage reciprocal rank (RR) as the evaluation metric and measure the ratio of instances where the first method outperforms (win) or achieves equal performance (tie) to the second method. The results are illus-

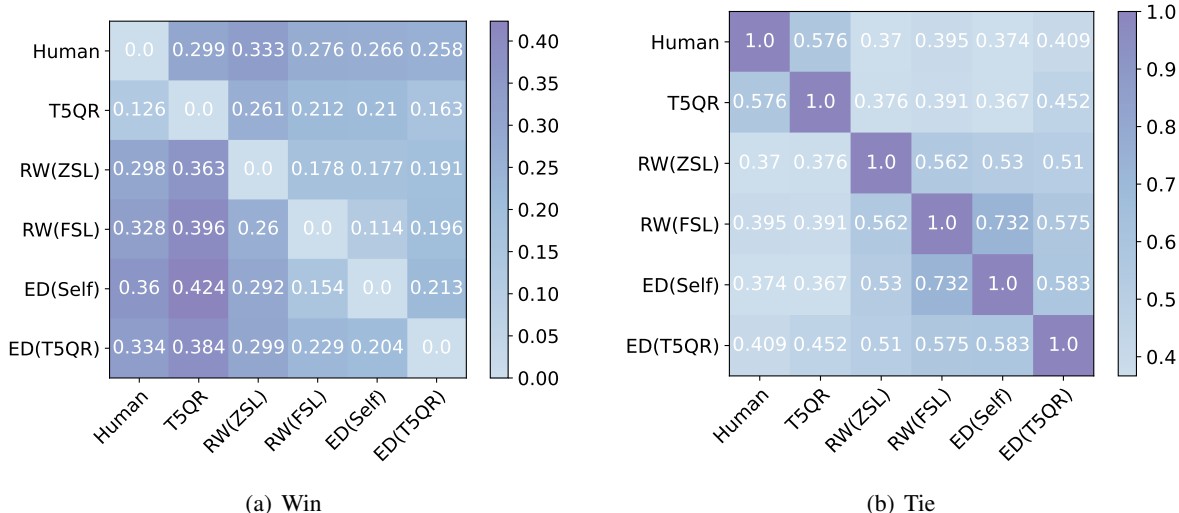

|  | (a) Win | (b) Tie |
|---|---|---|

Figure 4: The ratio of cases where query rewriting methods shown in the vertical axis achieve better performance (win) than or equal performance (tie) to methods shown in the horizontal axis in terms of the reciprocal rank (RR) metric on the overall QReCC test set. We adopt BM25 as the retriever in this study.

| | Query | QReCC | | | QuAC-Conv | | | NQ-Conv | | | TREC-Conv | | |
|---|---|---|---|---|---|---|---|---|---|---|---|---|---|
| | | NDCG@3 | R@5 | R@100 | NDCG@3 | R@5 | R@100 | NDCG@3 | R@5 | R@100 | NDCG@3 | R@5 | R@100 |
| **Sparse (BM25)** | Original | 7.98 | 12.06 | 28.63 | 8.02 | 12.10 | 27.70 | 7.80 | 11.14 | 29.06 | 7.96 | 14.82 | 42.99 |
| | Human | 35.97 | 50.71 | **98.48** | 36.57 | 51.21 | **98.35** | 36.58 | 51.65 | **98.96** | **23.15** | **38.54** | **98.92** |
| | T5QR | 30.13 | 43.09 | 85.66 | 30.55 | 43.57 | 85.86 | 30.47 | 42.86 | 84.42 | 21.39 | 35.58 | 87.06 |
| | ConQRR | - | - | 88.90 | - | - | 90.20 | - | - | 86.70 | - | - | 75.90 |
| | ConvGQR | 41.00 | - | 88.00 | - | - | - | - | - | - | - | - | - |
| | RW(ZSL) | 39.67 | 52.04 | 84.31 | 42.54 | 55.02 | 85.55 | 33.05 | 45.30 | 81.92 | 15.89 | 26.82 | 72.15 |
| | RW(FSL) | 43.82 | 56.60 | 88.34 | 46.83 | 59.59 | 89.86 | 37.61 | 50.85 | 86.56 | 16.00 | 27.36 | 69.05 |
| | ED(Self) | **46.43** | **58.86** | 88.15 | **50.20** | **62.55** | 89.95 | 37.94 | 50.89 | 85.72 | 14.32 | 26.28 | 66.49 |
| | ED(T5QR) | 44.85 | 57.69 | 89.91 | 47.80 | 60.49 | 91.09 | **38.77** | **51.88** | 87.97 | 17.71 | 32.08 | 77.18 |
| **Dense (GTR)** | Original | 10.90 | 15.23 | 27.22 | 10.09 | 14.38 | 25.64 | 12.00 | 15.97 | 29.26 | 20.51 | 27.09 | 46.50 |
| | Human | 39.42 | 54.85 | **86.86** | 36.68 | 52.44 | **86.46** | 50.89 | 65.24 | 88.54 | **42.01** | **56.06** | **87.24** |
| | T5QR | 34.37 | 48.19 | 79.85 | 32.06 | 46.09 | 79.65 | 43.98 | 56.94 | 79.92 | 36.79 | 50.40 | 83.20 |
| | ConQRR | - | - | 84.70 | - | - | 85.80 | - | - | 80.90 | - | - | 79.60 |
| | ConvGQR | 39.10 | - | 81.80 | - | - | - | - | - | - | - | - | - |
| | RW(ZSL) | 37.41 | 52.23 | 81.03 | 36.83 | 51.79 | 81.53 | 41.59 | 55.96 | 79.89 | 31.21 | 45.28 | 76.95 |
| | RW(FSL) | 40.62 | 56.12 | 84.76 | 40.17 | 55.85 | 85.33 | 45.41 | 60.19 | 84.67 | 29.87 | 44.83 | 75.34 |
| | ED(Self) | **41.80** | **57.11** | 85.61 | **42.06** | **57.50** | 86.42 | 44.08 | 59.27 | 84.42 | 28.44 | 41.87 | 76.15 |
| | ED(T5QR) | 41.49 | 56.46 | 85.91 | 41.02 | 56.11 | 86.25 | 46.39 | 60.27 | 85.35 | 30.56 | 47.71 | 82.12 |

Table 5: Passage retrieval performance of sparse and dense retrievers with various query rewriting methods on the QReCC test set and its three subsets. The best results are shown in bold and the second-best results are underlined.

trated in Figure 4. It can be seen that our proposed approaches, RW(FSL), ED(Self), and ED(T5QR), win more instances than human rewrites. For example, ED(Self) achieves higher performance in approximately 36% of instances, whereas human rewrites are better in only about 26.6% of instances. It can also be seen that T5QR outperforms human rewrites in only 12.6% of instances, again confirming that learning solely from human rewrites is not ideal. Moreover, we observe that ED(T5QR) outperforms T5QR in 38.4% of instances, while T5QR only wins in 16.3% of instances. This demonstrates that prompting LLMs as rewrite editors is effec-

tive in generating higher-quality query rewrites. Among the four variants of our proposed approach, we find that they share a relatively high tie ratio. For example, ED(Self) and RW(FSL) achieve equal performance in 73.2% of instances. This is reasonable since ED(Self) takes the rewrites produced by RW(FSL) as initial rewrites.

This analysis reveals that no single method possesses superior efficacy over all other methods, thereby indicating considerable potential for future research. It is crucial to emphasize that while we assert the limited informativeness of human rewrites in a general sense, it is possible that some

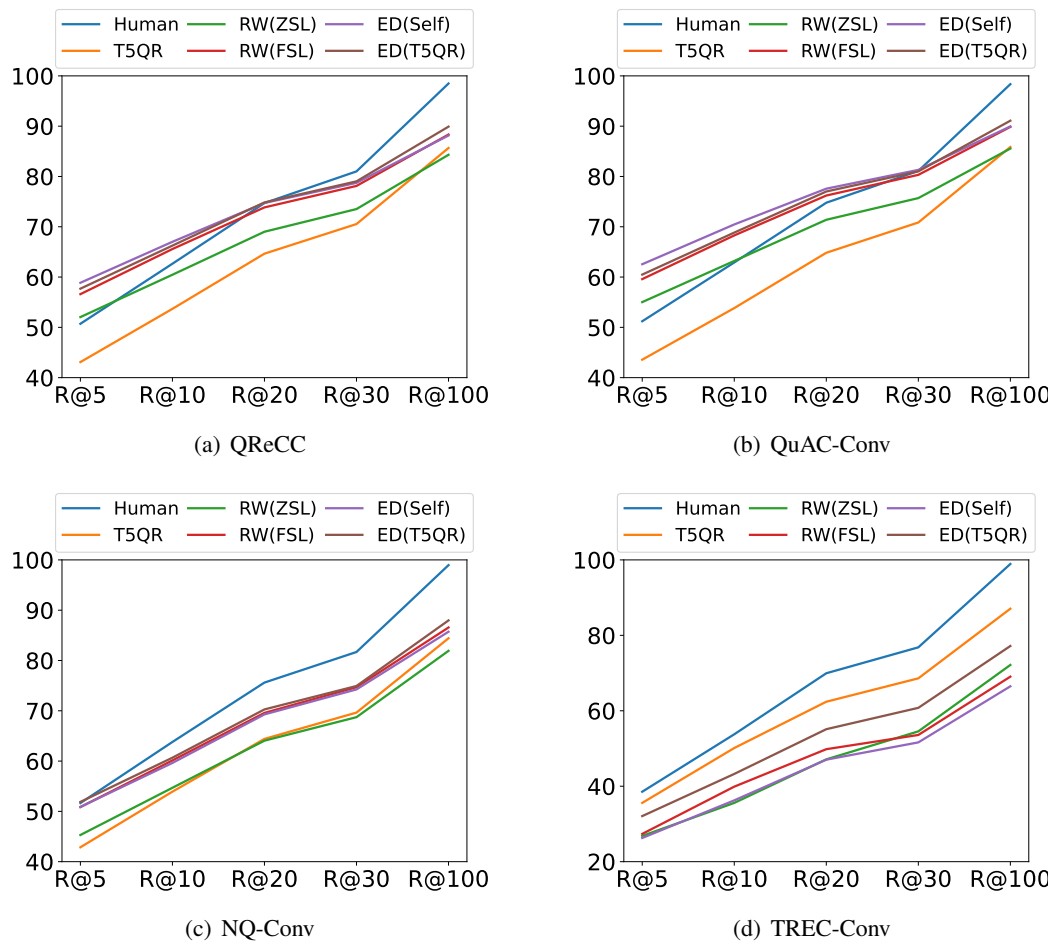

(a) QReCC

(b) QuAC-Conv

(c) NQ-Conv

(d) TREC-Conv

Figure 5: Recall value versus the cutoff rank $k \in \{5, 10, 20, 30, 100\}$, with BM25 as the retriever.

human rewrites are already informative enough.

## C.2 Main Results with Different Evaluation Metrics

Here, we expand the range of evaluation metrics to assess the retrieval performance more comprehensively. Specifically, we utilize normalized discounted cumulative gain with a cutoff rank of 3 (**NDCG@3**) and Recall@5 (**R@5**) to evaluate the relevance of top-ranked passages. Additionally, we employ Recall@100 (**R@100**) to consider the relevance of lower-ranked passages. The results are presented in Table 5. We find that our proposed approaches can substantially outperform human rewrites and all supervised baselines in terms of NDCG@3 and R@5 on the QReCC test set. For example, ED(Self) shows an absolute improvement of 10.46 in NDCG@3 compared to human rewrites. This finding indicates that our approaches can effectively rank relevant passages higher, which is particularly valuable for downstream tasks such as question answering, where only the top-ranked passages

are typically considered. Regarding R@100, we find that human rewrites consistently demonstrate the best performance. This is rational since human rewrites possess a higher guarantee of correctness. Although our proposed approaches are able to generate more informative query rewrites, they suffer from a lower guarantee of correctness as they are built upon LLMs. However, it is worth noting that our approach ED(T5QR) achieves the second-best results for both sparse and dense retrieval. Concerning the performance on the three subsets, our findings are similar to those presented in Section 5.1. Our approaches consistently achieve the best or second-best results in terms of NDCG@3 and R@5 on the QuAC-Conv and NQ-Conv subsets. However, they fall short of human rewrites and T5QR on the TREC-Conv subset. Nevertheless, our approach ED(T5QR) achieves higher performance than ConQRR, even though it employs reinforcement learning to optimize the model with retrieval performance as the reward.

We conduct further analysis on the trend of re-

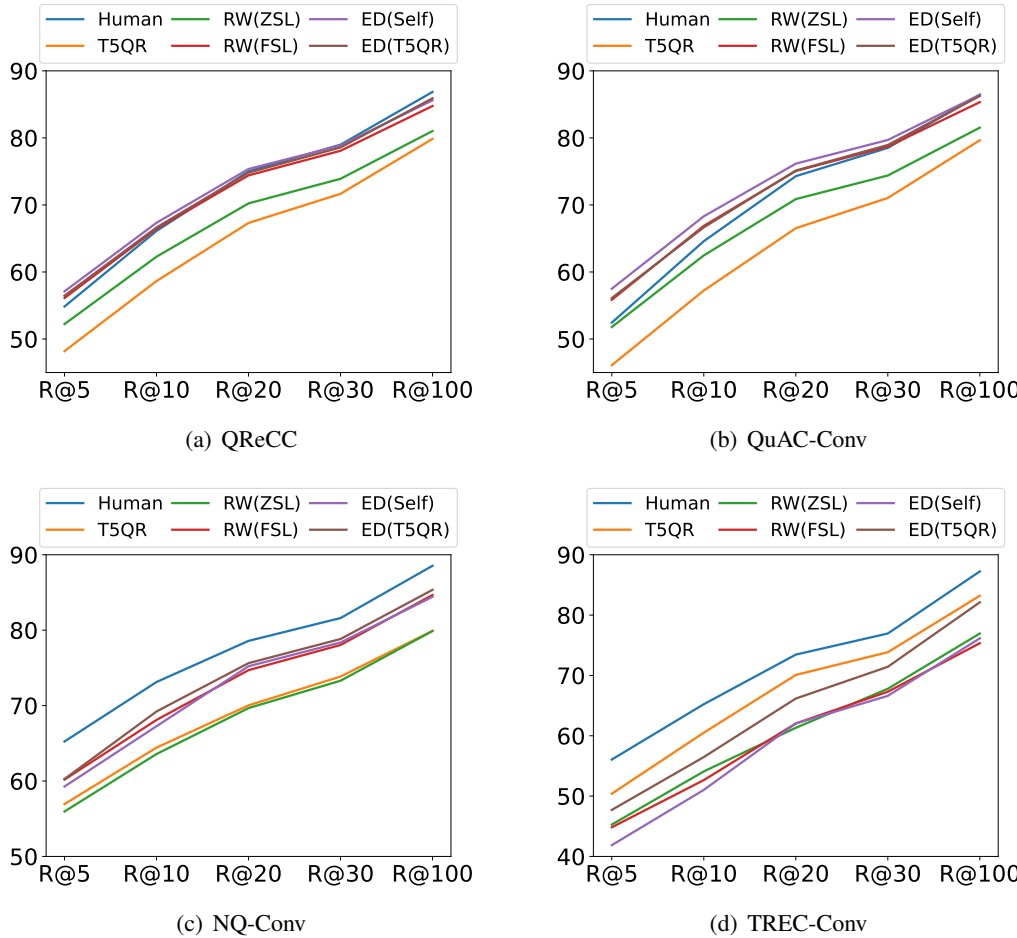

|  | (a) QReCC | (b) QuAC-Conv |
|--|-----------|---------------|
|  | (c) NQ-Conv | (d) TREC-Conv |

Figure 6: Recall value versus the cutoff rank $k \in \{5, 10, 20, 30, 100\}$, with GTR as the retriever.

call values by varying the cutoff rank $k$ within the range of $\{5, 10, 20, 30, 100\}$. Recall, which represents the fraction of relevant passages retrieved, is an important metric for evaluating the performance of retrieval results, particularly from a holistic point of view. The results of sparse retrieval and dense retrieval are illustrated in Figure 5 and Figure 6, respectively. As anticipated, increasing the value of $k$ leads to higher recall values across all methods, as a larger number of relevant passages are likely to be included in the results. Notably, our proposed approaches RW(FSL), ED(Self), and ED(T5QR) demonstrate commendable performance at different cutoff ranks (i.e., different values of $k$). They can surpass human rewrites when considering small values of $k$. However, they are inferior to human rewrites when $k$ becomes large. This observation suggests that the quality of rewrites generated by our approaches exhibits a higher variability. Even though many rewrites outperform human rewrites, there are also cases where the generated rewrites are significantly inferior. It is worthwhile to con-

| | Query | MRR | MAP | R@10 |
|---|---|---|---|---|
| **Sparse (BM25)** | RW(ZSL) | 42.63 | 41.31 | 60.46 |
| | -Informativeness | 41.52 | 40.20 | 59.60 |
| | -Correctness | 35.95 | 34.61 | 53.59 |
| | -Clarity | 33.43 | 32.16 | 50.87 |
| | -Nonredundancy | 36.58 | 35.23 | 55.02 |
| **Dense (GTR)** | RW(ZSL) | 40.64 | 38.95 | 62.28 |
| | -Informativeness | 39.94 | 38.28 | 61.44 |
| | -Correctness | 38.03 | 36.28 | 57.88 |
| | -Clarity | 35.78 | 34.11 | 54.82 |
| | -Nonredundancy | 38.53 | 36.82 | 59.27 |

Table 6: Ablation study by removing each of the four desirable properties from the instruction in RW(ZSL).

duct further research to optimize the worst-case performance of our proposed approach.

## C.3 Additional Ablation Study

In Table 6, we report the ablation results by removing each of the four desirable properties from the instruction in RW(ZSL). From the table, we can see

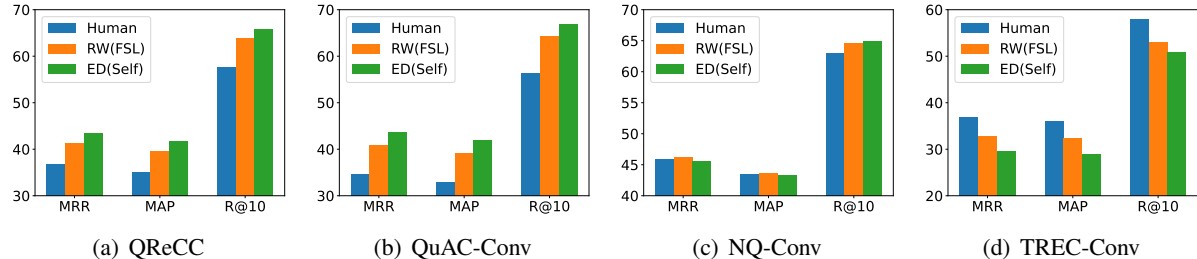

Figure 7: Distillation results of using GTR as the retriever. The legends indicate the sources of fine-tuning labels.

that all the four properties contribute to the performance improvement. Removing any one of them will lead to performance degradation. In particular, we can observe that removing either the correctness or clarity property leads to the most performance drop. This is because these two properties are crucial to ensure that the rewrites preserve the meaning of the original queries and are self-contained. Since the clarity requirement also contributes to the informativeness of rewritten queries, removing the informativeness requirement only seems to decrease the performance not that much.

In summary, this ablation study verifies that our proposed four properties in the instructions are essential to the success of prompting LLMs as query rewriters and informative query rewriting is critical for achieving better retrieval performance.

### C.4 Additional Distillation Results

The distillation results using GTR as the retrieval model are depicted in Figure 7. It can be observed that distillation outperforms using human rewrites as labels by a large margin on the QReCC test set. On the QuAC-Conv subset, distillation also consistently demonstrates superior performance. On the NQ-Conv subset, distillation surpasses the utilization of human rewrites as labels when RW(FSL) is employed as the teacher model.

Distillation offers benefits not only in enhancing retrieval performance but also in reducing time overhead. The rewriting latency comparison between T5QR and RW(FSL), with RW(FSL) serving as the teacher model, is presented in Table 7. The results indicate that the student model T5QR exhibits a significantly higher rewriting speed, approximately six times faster than RW(FSL).

### C.5 Case Study

We perform a case study to help understand the impact of informative query rewrites on retrieval performance more intuitively. We employ RW(FSL)

|  | T5QR | RW(FSL) |
|---|---|---|
| **Rewriting Latency** | 312 (ms/q) | 1867 (ms/q) |

Table 7: Comparison of rewriting latency in terms of milliseconds per query (ms/q). RW(FSL) is the teacher model, while T5QR is the student model.

as the query rewriter and BM25 as the retriever in this study. Table 8 showcases two successful examples. In the first example, adding the information "*during his time as Chancellor of the University of Chicago*" is crucial to understanding the original query comprehensively, thereby significantly improving the ranking of the relevant passage. In the second example, the added information exhibits a high degree of overlap with the gold passage, which also leads to better retrieval performance. Table 9 showcases one unsuccessful example. In this example, our approach RW(FSL) misinterprets the user query and expands it using wrong contextual information, resulting in worse retrieval performance.

### D Prompts

The prompts for utilizing LLMs as zero-shot and few-shot query rewriters and few-shot rewrite editors are presented in Table 10, Table 11, and Table 12, respectively.

**Conversational Context:** (id=511_4)

$Q_1$: What year was Robert Maynard Hutchins Chancellor of the University of Chicago?

$A_1$: Robert Maynard Hutchins served as University of Chicago's Chancellor from 1945 until 1951.

$Q_2$: Did Robert pull any sports out of the schools agenda?

$A_2$: Robert Maynard Hutchins eliminated the University of Chicago's football program, which he saw as a campus distraction.

$Q_3$: What collegiate conference of sports did he pull the university out of?

$A_3$: Robert Maynard Hutchins pulled the University of Chicago out of the Big Ten Conference.

**Current Question:**

$Q_4$: What degree did he make known for two year studies?

**Human Rewrite:**

$Q_4^*$: What degree did Robert Maynard Hutchins make known for two year studies? (**Rank=56**)

**Rewrite by RW(FSL):**

$Q_4'$: What degree program did Robert Maynard Hutchins make known for two year studies during his time as Chancellor of the University of Chicago? (**Rank=2**)

**Gold passage:**

. . . Hutchins was able to implement his ideas regarding a two-year, generalist bachelors during his tenure at Chicago, and subsequently had designated those studying in depth in a field as masters students . . .

---

**Conversational Context:** (id=1875_3)

$Q_1$: Where did Wu-Tang Clan's name come from?

$A_1$: Shaolin and Wu Tang is a film that inspired the name of the hip-hop group Wu-Tang Clan.

$Q_2$: When did the group form?

$A_2$: Wu-Tang Clan is an American hip hop group formed in the New York City borough of Staten Island in 1992.

**Current Question:**

$Q_3$: Who were the founding members?

**Human Rewrite:**

$Q_3^*$: Who were the founding members of Wu-Tang Clan? (**Rank=14**)

**Rewrite by RW(FSL):**

$Q_3'$: Who were the founding members of the American hip hop group Wu-Tang Clan, which was formed in the New York City borough of Staten Island in 1992 and named after the film Shaolin and Wu Tang? (**Rank=1**)

**Gold passage:**

. . . Wu-Tang Clan is an American hip hop group formed in the New York City borough of Staten Island in 1992, originally composed of rapper-producer RZA and rappers GZA, Ol´Dirty Bastard, Method Man, Raekwon, Ghostface Killah, Inspectah Deck, U-God and Masta Killa . . .

Table 8: Two successful examples showing that our proposed approach RW(FSL) can extract valuable contextual information to improve the informativeness of generated query rewrites. We highlight the additional information in generated rewrites compared to human rewrites in blue and the relevant parts in gold passages regarding the added information in red.

**Conversational Context:** (id=617_5)

$Q_1$: What did the singer, Cheryl do for the greater good?

$A_1$: In 2004, Cheryl's band, Girls Aloud released a cover of The Pretenders' I'll Stand by You as the official single for the BBC's charity telethon Children in Need.

$Q_2$: Did she do any other activities for charities?

$A_2$: In March 2009, Cheryl climbed Mount Kilimanjaro in aid of Comic Relief.

$Q_3$: Does she have any charities?

$A_3$: In February 2011, Cheryl launched her own charitable foundation with The Prince's Trust following a meeting with The Trust's President, HRH Charles, Prince of Wales.

$Q_4$: Are there any other interesting aspects about this article?

$A_4$: On 23 January 2015, Cheryl announced the launch of a second charity, once again alongside The Prince's Trust.

**Current Question:**

$Q_5$: For what charity?

**Human Rewrite:**

$Q_5^*$: What second charity did the singer, Cheryl launch in 2015? (**Rank=1**)

**Rewrite by RW(FSL):**

$Q_5'$: For which charity did Cheryl launch her own charitable foundation in February 2011, following a meeting with The Trust's President, HRH Charles, Prince of Wales? (**Rank=25**)

**Gold passage:**

... The charity was named Cheryl's Trust, and was set up with the aim of raising £2 million to build a centre, which will support up to 4000 disadvantaged young people in her native city of Newcastle. To raise these funds, Cheryl has thus far teamed up with Prizeo in March 2015 ...

Table 9: An unsuccessful example showing that our proposed approach RW(FSL) may misinterpret the original question and thus fail to identify truly relevant contextual information to augment the original query.

```
Given a question and its context, decontextualize the question by addressing
coreference and omission issues.  The resulting question should retain its
original meaning and be as informative as possible, and should not duplicate any
previously asked questions in the context.

Context: {Conversational context}
Question: {Question}
Rewrite:
```

Table 10: Prompt for utilizing LLMs as zero-shot query rewriters.

Given a question and its context, decontextualize the question by addressing coreference and omission issues. The resulting question should retain its original meaning and be as informative as possible, and should not duplicate any previously asked questions in the context.

Context: [Q: When was Born to Fly released?
A: Sara Evans's third studio album, Born to Fly, was released on October 10, 2000.
]
Question: Was Born to Fly well received by critics?
Rewrite: Was Born to Fly well received by critics?

Context: [Q: When was Keith Carradine born?
A: Keith Ian Carradine was born August 8, 1949.
Q: Is he married?
A: Keith Carradine married Sandra Will on February 6, 1982. ]
Question: Do they have any children?
Rewrite: Do Keith Carradine and Sandra Will have any children?

Context: [Q: Who proposed that atoms are the basic units of matter?
A: John Dalton proposed that each chemical element is composed of atoms of a single, unique type, and they can combine to form more complex structures called chemical compounds. ]
Question: How did the proposal come about?
Rewrite: How did John Dalton's proposal that each chemical element is composed of atoms of a single unique type, and they can combine to form more complex structures called chemical compounds come about?

Context: [Q: What is it called when two liquids separate?
A: Decantation is a process for the separation of mixtures of immiscible liquids or of a liquid and a solid mixture such as a suspension.
Q: How does the separation occur?
A: The layer closer to the top of the container-the less dense of the two liquids, or the liquid from which the precipitate or sediment has settled out-is poured off. ]
Question: Then what happens?
Rewrite: Then what happens after the layer closer to the top of the container is poured off with decantation?

Context: **{Conversational context}**
Question: **{Question}**
Rewrite:

Table 11: Prompt for utilizing LLMs as few-shot query rewriters.

Given a question and its context and a rewrite that decontextualizes the question, edit the rewrite to create a revised version that fully addresses coreferences and omissions in the question without changing the original meaning of the question but providing more information. The new rewrite should not duplicate any previously asked questions in the context. If there is no need to edit the rewrite, return the rewrite as-is.

Context: [Q: When was Born to Fly released?
A: Sara Evans's third studio album, Born to Fly, was released on October 10, 2000.
]
Question: Was Born to Fly well received by critics?
Rewrite: Was Born to Fly well received by critics?
Edit: Was Born to Fly well received by critics?

Context: [Q: When was Keith Carradine born?
A: Keith Ian Carradine was born August 8, 1949.
Q: Is he married?
A: Keith Carradine married Sandra Will on February 6, 1982. ]
Question: Do they have any children?
Rewrite: Does Keith Carradine have any children?
Edit: Do Keith Carradine and Sandra Will have any children?

Context: [Q: Who proposed that atoms are the basic units of matter?
A: John Dalton proposed that each chemical element is composed of atoms of a single, unique type, and they can combine to form more complex structures called chemical compounds. ]
Question: How did the proposal come about?
Rewrite: How did John Dalton's proposal come about?
Edit: How did John Dalton's proposal that each chemical element is composed of atoms of a single unique type, and they can combine to form more complex structures called chemical compounds come about?

Context: [Q: What is it called when two liquids separate?
A: Decantation is a process for the separation of mixtures of immiscible liquids or of a liquid and a solid mixture such as a suspension.
Q: How does the separation occur?
A: The layer closer to the top of the container-the less dense of the two liquids, or the liquid from which the precipitate or sediment has settled out-is poured off. ]
Question: Then what happens?
Rewrite: Then what happens after the layer closer to the top of the container is poured off?
Edit: Then what happens after the layer closer to the top of the container is poured off with decantation?

Context: **{Conversational context}**
Question: **{Question}**
Rewrite: **{Initial rewrite}**
Edit:

Table 12: Prompt for utilizing LLMs as few-shot editors.