# OpenReview forum: "Enhancing Conversational Search: Large Language Model-Aided Informative Query Rewriting"
_EMNLP/2023/Conference — EMNLP 2023 Findings_

### Official Review · Reviewer_ve5t · 2023-08-03

**Soundness:** 3

**Excitement:**

3: Ambivalent: It has merits (e.g., it reports state-of-the-art results, the idea is nice), but there are key weaknesses (e.g., it describes incremental work), and it can significantly benefit from another round of revision. However, I won't object to accepting it if my co-reviewers champion it.

**Paper Topic And Main Contributions:**

The authors in this paper propose an approach for conversation query rewriting. The QR task, differently from other works, focuses also on informativeness of the rewritten queries.

The proposed approach finds out that allowing the LLMs to rewrite then edits the queries. This strategy seems to perform best in terms of retrieval. Finally, the authors distill knowledge from LLMs into smaller LMs to decrease inference latency.

**Reasons To Accept:**

- Well written paper and an important task.

- Thorough evaluation and comparison against different competing baselines.

- Valuable insights on how to use LLMs to edit initial rewrites.

**Reasons To Reject:**

- The novelty of the paper is fairly limited to simply using LLMs to generate silver data, which are then used to train smaller LMs.



**Reproducibility:**

4: Could mostly reproduce the results, but there may be some variation because of sample variance or minor variations in their interpretation of the protocol or method.

**Reviewer Confidence:**

4: Quite sure. I tried to check the important points carefully. It's unlikely, though conceivable, that I missed something that should affect my ratings.

---

> ### Author Rebuttal · Authors · 2023-08-29
>
> Thank you very much for your insightful comments. Please find our responses below.
>
>  **Q1:** The novelty of the paper is fairly limited to simply using LLMs to generate silver data, which are then used to train smaller LMs.
>  > Thanks for raising this concern. We would like to clarify that **our main focus in this work is on generating informative query rewrites to address the limitation that human rewrites often lack enough informativeness**. To achieve this goal, we have designed clear and explicit instructions to prompt LLMs as query rewriters and rewrite editors. In addition, we have tested LLMs in both zero-shot and few-shot settings to generate informative query rewrites and observed that the generated query rewrites could achieve much higher performance than previous methods, even surpassing human rewrites. Considering that latency might be a concern in some application scenarios, we also investigated distilling the strong rewriting capabilities of LLMs into smaller models by treating the rewrites from LLMs as pseudo labels. It is worth emphasizing that **using LLMs to generate silver data to train smaller LMs is just one contribution among the manifold contributions we have made in this work**. We hope the reviewer could agree with us on this point. Moreover, we will release the generated rewrites to the community for future research, which would further strengthen our contributions. Our work also opens up many future directions, as shown in the response to Q2 of Reviewer uAia. We believe these strong points are sufficient to warrant the novelty of our work.

---

### Official Review · Reviewer_uAia · 2023-08-04

**Typos Grammar Style And Presentation Improvements:** none
**Soundness:** 3

**Excitement:**

3: Ambivalent: It has merits (e.g., it reports state-of-the-art results, the idea is nice), but there are key weaknesses (e.g., it describes incremental work), and it can significantly benefit from another round of revision. However, I won't object to accepting it if my co-reviewers champion it.

**Missing References:**

none

**Paper Topic And Main Contributions:**

This paper proposes using LLM to generate and edit query rewrites, addressing the limitation of human rewrites that lack contextual information for optimal retrieval in conversational search. Human rewriters focus on ambiguity but overlook contextual details from previous turns. Leveraging LLM's instruction-following abilities, the research defines desirable properties for query rewrites. However, LLMs may struggle with complex tasks, prompting the introduction of a "rewrite-then-edit" process, with initial rewrites generated by smaller models. To reduce memory and computational demands, a distillation method is suggested, fine-tuning smaller models with LLM's generated rewrites as training labels.

**Questions For The Authors:**

see above

**Reasons To Accept:**

- Comprehensive structure with comparisons against multiple baselines, including original queries, human rewrites, and query rewrites from other models.
- Introduction of the "rewrite editor" idea, leading to consistent and significant enhancement of retrieval performance on a specific dataset.
- Pioneering study defining four essential properties for instructing LLMs, contributing to the advancement of the field.
- Exploration of the applicability of using LLM's query rewrites to train smaller models through distillation.
- Design of experiments to measure the informativeness and correctness of query rewrites, providing valuable insights into the proposed approach's effectiveness.

**Reasons To Reject:**

- Further exploration is needed to determine the effect of the informativeness requirement in the instruction (i.e., the resulting question should be as informative as possible) on the generated query. For example, what is the difference between different informative requirements, such as "the resulting question should be as informative as it can be" or "the resulting question should be as helpful as possible"? Ablation studies show that even removing the informativeness requirement does not significantly decrease performance, which suggests that the effect of informativeness may not be useful. Additionally, although comparing whether including informativeness requirements in RW(ZSL) can show whether the informative requirement is effective, the results of ED(Self) should also be included since it is the main proposed method in the paper.
- In parts where the datasets NQ-Conv and TREC-Conv are used, the claim of emphasizing LLM rewrites over hunan rewrites seems to be undermined given the lower performances of all the rewrites that are not generated by humans (including LLMs and other baseline models). While the author explains that this is due to the obscure and self-contained characteristics of queries (particularly in TREC-Conv), the research does not continue to address LLMs’ potential applicability for such queries or point out future direction, which makes the original claim appear only effective under certain conditions.


**Reproducibility:**

3: Could reproduce the results with some difficulty. The settings of parameters are underspecified or subjectively determined; the training/evaluation data are not widely available.

**Reviewer Confidence:**

3: Pretty sure, but there's a chance I missed something. Although I have a good feel for this area in general, I did not carefully check the paper's details, e.g., the math, experimental design, or novelty.

---

> ### Author Rebuttal · Authors · 2023-08-29
>
> Thank you very much for your insightful comments. Please find our responses below.
>
> **Q1:** Further exploration is needed to determine the effect of the informativeness requirement in the instruction (i.e., the resulting question should be as informative as possible) on the generated query. For example, what is the difference between different informative requirements, such as "the resulting question should be as informative as it can be" or "the resulting question should be as helpful as possible"? Ablation studies show that even removing the informativeness requirement does not significantly decrease performance, which suggests that the effect of informativeness may not be useful. Additionally, although comparing whether including informativeness requirements in RW(ZSL) can show whether the informative requirement is effective, the results of ED(Self) should also be included since it is the main proposed method in the paper.
> > Thanks for the constructive suggestion. We agree that it can be useful to study the difference between different informative requirements. However, it is extremely challenging to measure the informativeness of a rewritten query and determine how much information a rewritten query should contain to achieve the best retrieval performance. The optimal amount of information in a rewritten query should be highly dependent on the query itself and the passages in the collection. For simplicity, we focus in this work on the requirement that the rewritten query should be as informative as it can be. In the Limitation section, we have pointed out that the most informative query rewrite may not be the best one for relevant passage retrieval, especially when passages in the collection are short. Informative query rewriting should prove more beneficial in the context of long passage or document retrieval. We agree that *''the rewritten query should be as helpful as possible''* tends to be a better informative requirement from the perspective of retrieval performance. However, it is challenging to ask an LLM to decide the helpfulness of a rewrite automatically, since the helpfulness can only be assessed after the retrieval process is finished. By comparison, it is much easier to instruct an LLM to generate as informative query rewrites as possible based on the conversational context. In addition, our experimental analyses showed that informative query rewriting could already lead to higher retrieval performance, even outperforming human rewrites.
> >
> > &nbsp;
> >
> >  Our ablation study showed that removing the informativeness requirement resulted in consistent performance degradation for both sparse and dense retrieval. We have also presented a case study in Appendix C.4, which shows more intuitively how an informative query rewrite can lead to higher retrieval performance. All these analyses validate the importance of the informativeness requirement. However, since we have considered four desirable properties of a well-crafted rewritten query in the instruction, it is reasonable that removing the informativeness requirement does not cause a drastic performance decrease. For example, the clarity property requires that the rewritten query should resolve all coreference and omission issues, which will also contribute to the informativeness of rewritten queries. If we remove the clarity requirement as well, the performance will drop further.
> >
> > &nbsp;
> >
> >  We conducted the ablation study using RW(ZSL) as the method because RW(ZSL) performs rewriting in the zero-shot setting, thus the performance will not be affected by the demonstrations used. It is worth mentioning that we only prompted LLMs as rewrite editors (for both ED(Self) and ED(T5QR)) in the few-shot setting with four demonstrations utilized. These demonstrations will inevitably interfere with the ablation study.
>
> **Q2:** In parts where the subdatasets NQ-Conv and TREC-Conv are used, the claim of emphasizing LLM rewrites over human rewrites seems to be undermined given the lower performances of all the rewrites that are not generated by humans (including LLMs and other baseline models). While the author explains that this is due to the obscure and self-contained characteristics of queries (particularly in TREC-Conv), the research does not continue to address LLMs’ potential applicability for such queries or point out the future direction, which makes the original claim appear only effective under certain conditions.
> > Thanks for this insightful comment. We have pointed out in our paper that it is crucial to take query characteristics into account when performing query rewriting with LLMs (#487-#490). Additionally, there can be many future directions. For example, we can train an auxiliary model to decide whether queries generated by LLMs should be preferred to those generated by models that have been trained on human rewrites. We can also combine human rewrites and LLM rewrites as labels through a proper weighting strategy to finetune query rewriting models. Furthermore, in this work, we have used a fixed set of demonstrations for all test queries. To achieve the best performance, it is essential to find appropriate demonstrations for each specific query. This would be an effective solution to tackle obscure or complicated queries. Another future direction can be parameter-efficient fine-tuning (e.g., LoRA [1]) of LLMs with retrieval performance as feedback. In this way, we will aim to optimize the helpfulness of rewritten queries rather than informativeness. We can add these potential future directions in the camera-ready version. Considering that each future direction requires dedicated research, we cannot address them in a single paper. However, given the excellent performance we have achieved and the thorough analyses we have performed, we believe we have made adequate contributions in this work. We also appreciate your comprehensive summary of our contributions in the Reasons to Accept section.
> >
> > &nbsp;
> >
> > [1] Hu, Edward J., et al. "Lora: Low-rank adaptation of large language models." arXiv preprint arXiv:2106.09685_ (2021).

---

### Official Review · Reviewer_H9aq · 2023-08-05

**Soundness:** 4

**Excitement:**

3: Ambivalent: It has merits (e.g., it reports state-of-the-art results, the idea is nice), but there are key weaknesses (e.g., it describes incremental work), and it can significantly benefit from another round of revision. However, I won't object to accepting it if my co-reviewers champion it.

**Missing References:**

Major related papers are cited.

**Paper Topic And Main Contributions:**

The paper proposes to use large language models for the task of query rewriting. Four dimensions about reformulations are defined to measure the quality of rewrites. Zero-shot setting and few-shot setting with GPT3.5 are evaluated. Beyond that, a “rewrite-then-edit” process is also evaluated where an initial query is also included during the rewiring process. The authors also evaluate the performance of distilled models using the output from LLMs. The experimental results show that their proposed models can outperform human performance on the QReCC dataset (and on most subsets). Quantitative analysis shows that their proposed methods are more informative and correct.

**Questions For The Authors:**

Line 413 mentions that “We disregard test instances without valid gold passage labels”. Do you mean using the initial contextualized query to obtain the candidates ? And is this step also applied in other related literature ?

**Reasons To Accept:**

- The capability of LLMs for zero/few shot rewriting is studied.
- The authors show that it is possible to distill small models from LLMs to achieve better performance than using the ground-truth labels on certain tasks.
- The paper is well-written and easy to follow.


**Reasons To Reject:**

- The novelty of the method is limited, considering Yu et al. SIGIR 2020, where GPT2 is used as the backbone.
- In Section 3.1, four dimensions are defined, while in the ablation study, only informativeness is considered.
- Only GPT3.5 is studied and the generalization ability is unknown, considering datasets used in the paper could be a part of the pretraining data of the LLMs.


**Reproducibility:**

5: Could easily reproduce the results.

**Reviewer Confidence:**

5: Positive that my evaluation is correct. I read the paper very carefully and I am very familiar with related work.

---

> ### Author Rebuttal · Authors · 2023-08-28
>
> Thank you very much for your insightful comments. Please find our responses below.
>
> **Q1:** The novelty of the method is limited, considering Yu et al. SIGIR 2020, where GPT2 is used as the backbone.
> > Thanks for raising this concern. We acknowledge that Yu et al. (SIGIR 2020) is an excellent work on conversational query rewriting. However, its focus is on leveraging ad hoc (non-conversational) search sessions to construct weak supervision data so as to mimic conversational search sessions. While this is a practical way to tackle the issue that human rewrites are expensive to collect, it cannot well address the problem of uninformative query rewriting, especially given that ad hoc search queries are usually short and contain only a few keywords. Besides, real conversational search sessions can be much more complicated than synthetic ones, which also challenges the effectiveness of models trained on these synthetic data.
> >
> >  &nbsp;
> >
> > In contrast, our work focuses on addressing the issue that human rewrites often lack sufficient information to achieve optimal retrieval performance, which is fundamentally different from Yu et al. (SIGIR 2020). Moreover, our contributions are manifold:
> > + We are the first to introduce the concept of informative conversational query rewriting.
> > + We have designed clear and explicit instructions to effectively prompt LLMs as query rewriters and rewrite editors to generate informative query rewrites, which not only solves the problem of high cost associated with human rewriting but also solves the problem that human rewrites are often uninformative enough.
> > + We have presented a comprehensive set of experimental analyses. Impressively, our results show that the generated informative query rewrites can achieve better retrieval performance than human rewrites.
> >
> > We believe these contributions are sufficient to warrant the novelty of our work. We will also release the generated rewrites for future research. Given the limitations of using human rewrites as ground-truth labels, we argue that it does not make much sense to design and train new models to mimic human rewrites. We hope our work will provide valuable insights to the community.
>
> **Q2:** In Section 3.1, four dimensions are defined, while in the ablation study, only informativeness is considered.
> > Thanks for pointing out this issue. We investigate only informativeness in the ablation study for two reasons. Firstly, our main focus in this work is on the informativeness of rewritten queries. The other three dimensions are introduced to ensure that the generated rewritten queries are as accurate as possible (i.e., retain their original meanings and intents). Our motivation is that only when the rewrites remain correct can it make sense to study the impacts of informativeness. Secondly, since we take ChatGPT as the LLM in this work, it is costly to explore the impacts of each dimension. Nonetheless, we have quantitatively analyzed the correctness and clarity of generated rewritten queries in Section 5.2 and Table 2.
>
> **Q3:** Only GPT3.5 is studied and the generalization ability is unknown, considering datasets used in the paper could be a part of the pretraining data of the LLMs.
> > Thanks for this suggestive comment. We agree that it could be a limitation to experiment with only one LLM, i.e., ChatGPT. We have mentioned this point in the Limitation section. Our primary concern is that different LLMs have different capabilities of following instructions to perform rewriting. Other (smaller) LLMs may not achieve the same level of performance as ChatGPT. In other words, the generalization ability is related to the basic capacities of LLMs. However, we do not think that data contamination could affect the generalization ability significantly. It is worth emphasizing that our goal is to prompt LLMs to generate rewrites that are more informative than human rewrites. These informative rewrites are not included in the original dataset. Therefore, even if data contamination happens, LLMs would still struggle to generate informative query rewrites without our proposed instruction and the adopted demonstrations. In fact, we found in our initial trials that even ChatGPT showed inferior performance to human rewrites when we simply set the instruction to *''rewrite the question into a self-contained and contextually independent one''*. This further verifies that the performance improvement is not caused by data contamination.
>
> **Q4:** Line 413 mentions that “We disregard test instances without valid gold passage labels”. Do you mean using the initial contextualized query to obtain the candidates? And is this step also applied in other related literature?
> > Thanks for the question. In our experiments, we removed instances that do not have valid gold passage labels from the test test. This is because when no gold passage labels are provided, it is challenging to evaluate whether the retrieved passages are truly relevant or not. The gold passage labels are provided in the dataset. We do not obtain the labels by ourselves. This preprocessing strategy has also been adopted in ConQRR (refer to footnote 8) [1].
> >
> >  &nbsp;
> >
> > [1] Wu, Zeqiu, et al. "ConQRR: Conversational Query Rewriting for Retrieval with Reinforcement Learning." EMNLP 2022, https://aclanthology.org/2022.emnlp-main.679.pdf.

---

### Meta-Review · Area_Chair_frqq · 2023-09-17

**Recommendation:** 3

**Metareview:**

This paper presents a new approach utilizing LLMs for the generation and refinement of query rewrites, addressing a critical limitation inherent in human-generated rewrites – the absence of contextual information required for optimal retrieval in conversational search. Leveraging the instruction-following capabilities of LLMs, this research delineates a set of highly desirable attributes for query rewrites.
Considering that LLMs can face challenges when confronted with intricate tasks, this paper also introduces a "rewrite-then-edit" process.

Although it is essential to underscore the potential impact of this method on the field of conversational search, some reviewers have expressed concerns about the paper's novelty.

---

### Decision · Program_Chairs · 2023-10-07

**Decision:**

Accept-Findings

**Comment:**

This paper presents a new approach utilizing LLMs for the generation and refinement of query rewrites, addressing a critical limitation inherent in human-generated rewrites – the absence of contextual information required for optimal retrieval in conversational search. Leveraging the instruction-following capabilities of LLMs, this research delineates a set of highly desirable attributes for query rewrites.
Considering that LLMs can face challenges when confronted with intricate tasks, this paper also introduces a "rewrite-then-edit" process.

Although it is essential to underscore the potential impact of this method on the field of conversational search, some reviewers have expressed concerns about the paper's novelty.